# A Lesson from a Measles Outbreak among Healthcare Workers in a Single Hospital in South Korea: The Importance of Knowing the Prevalence of Susceptibility

**DOI:** 10.3390/vaccines11091505

**Published:** 2023-09-20

**Authors:** Sungim Choi, Jae-Woo Chung, Yun Jung Chang, Eun Jung Lim, Sun Hee Moon, Han Ho Do, Jeong Hun Lee, Sung-Min Cho, Bum Sun Kwon, Yoon-Seok Chung, Seong Yeon Park

**Affiliations:** 1Department of Infectious Diseases, Dongguk University Ilsan Hospital, College of Medicine, Dongguk University, Goyang-si 10326, Gyeonggi-do, Republic of Korea; drsichoi87@gmail.com; 2Department of Infection Control, Dongguk University Ilsan Hospital, College of Medicine, Dongguk University, Goyang-si 10326, Gyeonggi-do, Republic of Korea; 3Department of Laboratory Medicine, Dongguk University Ilsan Hospital, College of Medicine, Dongguk University, Goyang-si 10326, Gyeonggi-do, Republic of Korea; 4Department of Emergency Medicine, Dongguk University Ilsan Hospital, College of Medicine, Dongguk University, Goyang-si 10326, Gyeonggi-do, Republic of Korea; 5Department of Pediatrics, Dongguk University Ilsan Hospital, College of Medicine, Dongguk University, Goyang-si 10326, Gyeonggi-do, Republic of Korea; 6Department of Rehabilitation Medicine, Dongguk University Ilsan Hospital, College of Medicine, Dongguk University, Goyang-si 10326, Gyeonggi-do, Republic of Korea; 7Division of Infectious Disease Diagnosis Control, Honam Regional Center for Disease Control and Prevention, Korea Disease Control and Prevention Agency, Gwangju 61947, Gyeonggi-do, Republic of Korea

**Keywords:** measles, outbreak, HCWs, vaccination, immunity

## Abstract

Background: Despite the high vaccination coverage rate, in-hospital transmission of measles continues to occur in South Korea. We present a measles outbreak in which two healthcare workers (HCWs) with presumptive evidence of measles immunity were infected by a patient with typical measles at a single hospital in South Korea. This facilitated the evaluation of measles seroprevalence in all HCWs. Methods: In 2018, suspected patients and contacts exposed during a measles outbreak were investigated based on their medical histories and vaccination status. Cases were confirmed by the detection of measles-specific immunoglobulin M or RNA. After the measles outbreak in 2018, measles IgG testing was conducted on a total of 972 HCWs for point-prevalence, including those exposed to the measles. In addition, we have routinely performed measles IgG tests on newly employed HCWs within one week of their hire date since 2019. The measles vaccine was administered to HCWs who tested negative or equivocally negative for IgG antibodies. Results: An index patient who returned from China with fever and rash was diagnosed with measles at a hospital in Korea. Two additional HCWs were revealed as measles cases: one was vaccinated with the two-dose measles–mumps–rubella (MMR) vaccine, and the other, who was born in 1967, was presumed to have immunity from natural infection in South Korea. All three patients harbored the same D8 genotype. No additional measles cases were identified among the 964 contacts of secondary patients. A total of 2310 HCWs, including those tested during the 2018 outbreak, underwent measles IgG tests. The average age at the time of the test was 32.6 years, and 74.3% were female. The overall seropositivity of measles was 88.9% (95% confidence interval, 87.5–90.1). Although the birth cohorts between 1985 and 1994 were presumed to have received the measles–rubella (MR) catch-up vaccination in 2001, 175 (89.3%) HCWs were born after 1985 among the 195 seronegative cases. Conclusion: Despite high population immunity, imported measles transmission occurred among HCWs with presumed immunity. This report underscores the importance of understanding the prevalence of measles susceptibility among newly employed HCWs. This is important for policymaking regarding hospital-wide vaccinations to prevent the spread of vaccine-preventable diseases.

## 1. Introduction

Measles is an acute infectious disease caused by the measles virus and is one of the most contagious diseases. In a 100% susceptible population, a single measles case results in 12 to 18 secondary cases on average [1], which is one of the highest for any human pathogen. Measles are spread by direct contact with droplets from the respiratory secretions of infected people and via the airborne route. Patients with measles are infectious from 4 days before to 4 days after their rash onset. The fact that the measles virus is contagious before the onset of recognizable symptoms can hinder quarantine measures’ efficacy, although isolation of susceptible contacts is recommended.

In South Korea, the measles-containing vaccine (MCV) became available in 1965, and the trivalent measles–mumps–rubella (MMR) vaccine was introduced in the early 1980s. Two doses of the MMR vaccine were implemented in 1997 [2], with the first dose being given at 12–15 months of age and the second dose at 4–6 years of age. Nonetheless, until 2001, nationwide episodes of measles outbreaks occurred at intervals of approximately 4 to 6 years. Following the catch-up vaccination initiative targeting school-age children in 2001, measles incidence saw a significant reduction to fewer than 1 case per 100,000 members of the population [3]. In March 2014, the World Health Organization (WHO) verified that measles was eliminated in South Korea, as a result of a high-quality case-based surveillance system and population immunity, achieved by a high vaccination rate (>95.0% since 1996) [4,5].

The Korea Disease Control and Prevention Agency (KDCA) criteria for acceptable presumptive evidence of immunity against measles include at least one of the following: written documentation of the two-dose MMR vaccination, laboratory evidence of immunity, laboratory confirmation of measles, or proof of birth before 1967 [3]. Despite these criteria for immunity against measles and decades of vaccine use, measles imports and limited local or in-hospital transmission continue in South Korea [6,7,8].

Here we are presenting a measles outbreak in a single hospital in South Korea, where an imported patient with typical measles subsequently transmitted the measles infection to two healthcare workers (HCWs), with presumptive evidence of measles immunity. In response, our hospital conducted a serological survey of measles in all HCWs and offered free MMR vaccines to seronegative HCWs. In addition, we have routinely performed measles antibody tests on newly hired HCWs since 2019. We also present measles seroprevalence data for HCWs stratified by birth year. This is to establish an infection control and prevention strategy for measles that is applicable to hospitals in countries with a measles-eliminated status.

## 2. Methods

We investigated the measles seroprevalence of HCWs at a single hospital in South Korea with a total of 2310 HCWs from 2018 to 2022. After an outbreak of measles in 2018, our hospital performed measles IgG tests for all HCWs (*n* = 972) for point-prevalence, including those exposed to the measles outbreak. In addition, we have routinely performed measles antibody test for new HCWs within one week of their hire date since 2019 (*n* = 1338).

### 2.1. Case Definition

Clinical measles virus infection was defined as fever and a typical maculopapular rash of measles [6]. Cases were confirmed by laboratory tests. A laboratory-confirmed case was defined as a clinical case patient with one or more of the following results: presence of measles immunoglobulin M (IgM) antibody or measles-specific RNA in a nasopharyngeal swab.

### 2.2. Laboratory Testing

Serum specimens were tested for measles-specific IgM antibodies using an IgM capture enzyme immunoassay (EIA), as described previously [9]. Measles-specific IgG was tested by an enzyme-linked immunoassay (ELISA) for measles IgG (EUROIMMUN, Lübeck, Germany), and the test results were interpreted according to the manufacturer’s instructions. IgM/IgG index ratios were derived by dividing the net absorbance values measured for IgM by those for IgG [10]. The IgM/IgG ratios were compared as a measure of primary vs. secondary immune responses to infection. Index ratios > 1 suggested a primary immune response to measles, whereas ratios < 1 indicated a secondary response [10].

Nasopharyngeal swabs were sent to public health laboratories for polymerase chain reaction (PCR). Reverse transcription polymerase chain reaction (RT-PCR) and genotyping were performed.

### 2.3. Real-Time RT-PCR for Measles Virus (MeV) Detection

Real-time RT-PCR assays were performed to detect the measles virus N gene using a 7500 Fast Real-time PCR System (Applied Biosystems). Forward (MVN1139-F:5′-TGGCATCTGAACTCGGTATCAC-3′) and reverse (MVN1213-R:5′-TGTCCTCAGTAGTATGCATTGCAA-3′) primers were used. A probe (MVNP1163-P:5′-CCGAGGATGCAAGGCTTGTTTCAGA-3′) was labeled at the 5′ terminus with a fluorescent reporter dye, 6-carboxyfluorescein (FAM), and at the 3′ terminus with a non-fluorescent quencher, black hole quencher-1 (BHQ1). The amplification conditions were as follows: 50 °C for 30 min, followed by 95 °C for 10 min, and 40 cycles of 95 °C for 15 s and 60 °C for 1 min. Real-time RT-PCR assays were verified by the Standard Operation Protocol Verification Committee of the KDCA.

### 2.4. RNA Extraction, RT-PCR, and Sequencing

Viral RNA was extracted from throat swab samples or infected cell supernatants using a QIAamp Viral RNA Mini Kit (Qiagen, Venlo, the Netherlands) according to the manufacturer’s instructions. The highly variable 450-nucleotide (nt) region in the carboxy-terminus of the nucleocapsid protein (N-450) was amplified and sequenced for genotyping using forward (MeV216:5′-TGGAGCTATGCCATGGGAGT-3′) and reverse (MeV214:5′-TAACAATGATGGAGGGTAGG-3′) primers. RT-PCR was performed using a OneStep RT-PCR Kit (Qiagen, Venlo, The Netherlands) in accordance with the manufacturer’s instructions. The amplification conditions were as follows: 50 °C for 30 min, followed by 95 °C for 15 min, and 40 cycles of 95 °C for 30 s, 95 °C for 30 s, and 72 °C for 30 s, with a final 10 min extension at 72 °C.

### 2.5. Phylogenetic Analysis

The sequences obtained herein were aligned with the CLC Main Workbench 7.9.1, including all genotypes reference sequences from GenBank. MEGA X was used to generate phylogenetic trees through the neighbor-joining method using the maximum composite likelihood-parameter distance matrix listed in the software; bootstrap values were obtained through random sampling of 1000 replicates.

## 3. Results

### 3.1. Outbreak Presentation

A 41-year-old man with fever, cough, and rash visited the emergency department of another hospital on 17 May 2018 (Case 1). He lived in China, where there was a measles outbreak at the time [11]. When his symptoms did not improve, he visited the emergency room (ER) of our hospital. The patient had conjunctivitis, pharyngeal infection, and a maculopapular rash that started on his face and spread to his chest 2 days after fever onset. The primary doctor initially diagnosed the patient with viral exanthema and placed him in the ER Ward 1. However, after a medical examination by a senior doctor, the patient was clinically diagnosed with measles. He was sent to an isolation room in the ER after spending 4 h in Ward 1. He was admitted to an isolation room in the general ward, and the PCR results indicated he was measles-virus-positive 7 days after admission. Measles IgM and IgG were tested positive and borderline, respectively.

The ER has three wards and two isolation rooms. The three wards were adjacent and separated by full-height walls, and each of the wards are designed to be an open-configuration without doors. Each ward has eight beds separated by curtains.

The infection control team performed contact tracing based on exposure and immune status [12]. Initially, contacts were identified as those who had face-to-face contact or spent at least 15 min in Ward 1 with the index case. In total, there were 30 exposed patients and 5 exposed HCWs. All HCWs were considered measles-immune due to their two-dose MMR vaccination documentation, and there were no susceptible high-risk contacts among the 30 exposed patients.

Eleven days after the index patient visited the ER, an ER HCW developed a mild fever. Three days later, a rash appeared on her face and spread to her trunk. While this HCW had no prodromal symptoms, such as cough, coryza, or conjunctivitis, both PCR, and measles IgM and IgG serology produced positive results (Case 2). The HCW had a record of the two-dose MMR vaccination. In addition, the HCW was not initially identified as having contact with the index case. Further investigation showed that the HCW had no face-to-face contact and was not in Ward 1 for more than 15 min; however, the HCW worked in the ER at the same time for 4 h. Case 2 was considered infectious 4 days before the rash onset. There were 158 exposures in HCWs and 206 exposures in patients. After Case 2, the definition of contact was modified. Contacts were considered all patients and HCWs who stayed in the ER, not just Ward 1, from the time the index patient had entered the ER to an additional two hours after he exited the ER. The exposed HCWs had their immunity against measles checked, including their two-dose MMR vaccination status and birth year. We provided MMR vaccination to exposed HCWs born after 1967 who did not document two-dose MMR vaccination. However, after Case 2 occurred, we had to evaluate measles IgG in the ER’s HCWs, regardless of birth year (age).

Another HCW in the ER, born in 1967, developed two macular rashes on the neck 17 days after exposure to the index patient (Case 3). This patient also did not exhibit any prodromal symptoms. The patient was a nursing assistant who was not involved in the treatment of the index case. Although Case 3 was not in Ward 1 for more than 15 min, she worked in the ER for 4 h with the index case present. A nasopharyngeal swab for PCR was performed, but an IgG/IgM test was not performed because her symptoms were atypical. However, the HCW’s PCR results came back positive for the measles virus, resulting in 135 HCWs and 316 patients being exposed.

We performed phylogenetic analysis and found that the genotype of the measles virus in all three cases was D8, which circulates in China (Figure 1) [13,14]. This outbreak resulted in 3 cases and 964 potential exposures, with 341 HCWs and 623 ER patients. The hospital infection control team held an urgent meeting with the ER and local public health control department to coordinate further actions. We compiled a list of exposed HCWs and admitted patients and implemented measures to prevent further infection. The measles IgG test was performed on all HCWs who entered and exited the ER, regardless of measles exposure or MMR vaccination status. Among the 341 HCWs, 68 were seronegative for measles IgG. Thirty-five were considered contacts and received an MMR vaccination. They were quarantined for three weeks after the last exposure. The other 33 received an MMR vaccination but were not quarantined. Local public health authorities contacted patients discharged from the hospital after exposure to measles in the ER, monitored them for symptoms, and administered vaccinations within 72 h. The hospital infection control team has established a special isolated outpatient clinic for patients diagnosed with measles. In the following week, two suspected measles cases among the contacts visited an isolated outpatient clinic. Both individuals were ER inpatients when Cases 2 and 3 were admitted and infectious. Both nasopharyngeal swabs tested positive for measles RNA; however, the virus was shown to belong to the A genotype, which is a vaccine strain. No additional cases were observed among the 964 contacts of secondary patients.

### 3.2. Seroprevalence of Measles in All HCWs

Following the measles outbreak, our hospital has implemented routine measles antibody testing for newly hired HCWs since 2019, regardless of birth year or MMR vaccination status. Including HCWs tested at the time of the outbreak in 2018, 2310 HCWs had undergone measles IgG testing by 2022. Table 1 shows the seroprevalence data for measles for HCWs stratified by year of measles IgG testing. Figure 2 shows the seroprevalence data stratified by birth year.

The average age of those tested was 32.6 years, and 74.3% were female. The overall measles seropositivity was 88.9% (95% confidence interval, 87.5–90.1). HCWs born up to 1967 had 100.0% seropositivity, indicating full herd immunity. However, HCWs born in 1974 had a lower seropositivity of 84.2% (16/19), suggesting that age alone cannot reliably determine immunity, even though this interpretation is limited by a small number. Among the 195 seronegative cases, 89.3% (175/195) HCWs were born after 1985, despite the presumption that birth cohorts between 1985 and 1994 received the measles–rubella (MR) catch-up vaccination in 2001 [15]. Notably, the birth cohort between 1994 and 1996 had a substantially low seropositivity for measles, thus signifying pockets of under-immunity. The measles vaccine was administered to HCWs who tested negative or equivocally negative for IgG antibodies.

## 4. Discussion

Following the measles outbreak in 2018, we made significant changes to our measles infection control policies. First, we developed a procedure for immediately isolating and testing patients suspected of measles and reporting them to the infection control office. Second, we conducted measles IgG tests for all HCWs, including those with administrative roles, regardless of their MMR vaccination status or birth year. Third, we now perform measles IgG tests on new HCWs regardless of their MMR vaccination status and age. In addition, we offer the MMR vaccine to those with negative test results.

To prevent nosocomial transmission, pre-emptive outpatient triage and in-hospital isolation with airborne transmission precautions should be applied to patients with suspected measles before confirming the diagnosis. Therefore, early suspicion and recognition are crucial for preventing widespread exposure and subsequent outbreaks. However, measles has been eliminated and is not endemic in South Korea [4], and young doctors may lack familiarity with the clinical manifestations of measles due to limited local transmission. Furthermore, during this outbreak, the index case exhibited typical symptoms, whereas Cases 2 and 3 exhibited mild and atypical symptoms. Patients with a measles infection in high-vaccination-rate countries may present with modified measles, posing another challenge for early diagnosis [16]. In a measles outbreak among young adults in Victoria in 2002, all but 1 of the 22 hospitalized cases experienced delayed diagnoses due to the low index of suspicion, presenting opportunities for further transmission within the healthcare setting [17]. To address this concern, we took steps to educate medical residents about the presentations of atypical or modified measles. Additionally, we implemented a rapid isolation protocol for patients suspected of having measles in order to mitigate the risk of transmission.

Failure to initially identify Cases 2 and 3 as contacts of Case 1 led to delayed recognition of the disease. The accurate identification of all contacts, including transient or brief contacts, is crucial to avoid such delays and ensure timely prophylaxis. HCWs have an 18.6-fold higher risk of acquiring measles compared to adults in the community [18] and are at high risk of transmitting measles to vulnerable groups. However, identifying all HCWs can be challenging because of the multiple brief and undocumented encounters. Additional case finding and a more comprehensive approach to contact tracing are necessary.

Documented administration of the two-dose MMR vaccine is generally considered to provide long-term protection against measles and evidence of measles immunity regardless of the serological test results; however, caution for durability of protective immunity should be exercised, especially in young Korean HCWs. Seropositivity for measles gradually decreases over time after the second dose of the MMR vaccine in the Korean population, as reported by Kang et al. [19]. Birth cohorts after 1994 in South Korea received routine second doses of the MMR vaccine with a 95% coverage rate by submitting vaccination certificates before entering elementary school, but they have the lowest measles seropositivity at about 40%. Kim et al. reported similar results among HCWs: the 1994 birth cohort in their hospital had the lowest antibody titers (approximately 30%), followed by the earlier cohort [20]. In our study results, routine measles antibody testing for newly hired HCWs over several years revealed that 11.1% of all tested individuals were susceptible to measles. Among them, 89.3% were born after 1985 and were expected to have received a two-dose MMR vaccine through national and catch-up vaccination programs in 2001. The seropositivity of HCWs born after 1994 was as low as 84.1% (745/886), and in particular, the seropositivity of HCWs born in 1995 was the lowest at 76.2% (138/181). Case 2, born in 1995, received a two-dose MMR vaccine and tested positive for measles IgG at the onset of the rash. Previous studies on measles outbreaks in highly vaccinated populations have suggested potential factors for vaccine failure, such as waning immunity [21,22,23]. Similar to our current study, in one retrospective study conducted in 2022 in Taiwan, which declared measles elimination in 2007 subsequent to the introduction of the measles vaccine in 1978, anti-measles IgG-positive rates were investigated among HCWs born before 1977 and after 1978. The seropositive rates for these two groups were found to be 94.8% and 70.2%, respectively (*p* < 0.001), strongly associated with prior exposure to natural measles infection [24].

The factors underlying the discrepancy between the low and high seropositive rates of the two-dose vaccination remain unclear. One potential explanation could be primary vaccine failure (failure to seroconvert after vaccination), which was considered less significant because primary vaccine failure was deemed less significant in the two-dose vaccination strategy era. Secondary vaccine failure (waning immunity after seroconversion) following a two-dose vaccination is a probable cause. In South Korea, administering the first vaccine dose at 12 months of age is common, and factors such as interference with maternal antibodies and an immature immune response to the initial dose may contribute to waning immunity in individuals relying solely on vaccine-induced protection without natural exposure to measles [25,26]. Also, vaccine-induced immunity might attenuate swiftly due to the diminishing exogenous boosting effect derived from encounters with circulating measles virus, a decline concomitant with reduced measles incidence. Measles avidity assays may provide useful information for assessing the occurrence of measles in highly vaccinated populations [27,28].

However, age alone also does not ensure immunity against measles in healthcare settings. In the United States, most people born before 1957 are presumed immune due to natural infection [29]. In South Korea, the 30–34 age group showed 95.4% antibody positivity in a 2002 immunity survey [30], while the 1965–1974 birth cohort had 97% measles immunity in 2014 [19,31]. Thus, being born before 1967 is considered presumptive evidence of measles immunity in South Korea. Although the measles IgG status of Case 3 was not confirmed, her atypical presentation suggested partial immunity. The KCDC’s 2012 adult immunization guidelines recommended measles vaccination for medical personnel born after 1967 without evidence of immunity [32]. However, owing to frequent measles outbreaks in healthcare facilities, the guidelines were revised in May 2019 to advise two doses of MMR vaccination for those at risk of contact with patients with measles or working in high-risk medical institutions without evidence of immunity [32].

Fortunately, all the patients with measles recovered without complications, and despite over 900 identified exposures, no additional cases were found. While we did not assess the baseline immune status of Cases 2 and 3 against measles, our findings align with those of other studies, indicating that documented cases of secondary vaccine failure are unlikely to transmit the virus [27,28]. It is possible that the neutralizing antibody levels declined enough in secondary patients to permit symptomatic infection, but a robust memory response upon re-exposure likely shortened their infectious period. Therefore, it can be inferred that both HCWs were partially immune to measles.

This observation emphasizes the significance of identifying the prevalence of measles susceptibility among newly hired HCWs, regardless of age, to guide hospital-wide vaccination policies to prevent vaccine-preventable diseases. Young Korean HCWs should receive the highest priority for enhancing herd immunity in hospitals.

Our study had certain limitations that should be acknowledged. First, although Case 2 tested positive for measles IgG during diagnosis, we were unable to confirm secondary vaccine failure due to the lack of measles IgG avidity testing, such as plaque reduction-neutralization (PRN) tests. Additionally, measles serology was not performed in Case 3. As a result, we could not determine the immune status baseline of Cases 2 and 3 before measles infection. Second, the National Immunization Registry Information System, launched in 2000, was not widely used until 2011; therefore, MCV immunization records may not be complete for some HCWs. Previous studies examining the seroprevalence of highly contagious diseases among HCWs, which relied on self-reported medical history, have revealed the limitations of this approach in accurately confirming immunity on account of recall bias [33,34,35]. Lastly, this was a single-center study, and the seropositivity of young HCWs was relatively high compared with other studies in South Korea. However, this study provides valuable seroepidemiological data for establishing hospital vaccination policies owing to the large cohort of HCWs.

In conclusion, despite high MMR vaccination rates in South Korea, measles importation and limited local transmission continue to occur. As the WHO declared the end of the COVID-19 global health emergency in May 2023, it is anticipated that there will be a rise in the importation of measles cases as well as an overall upsurge in measles outbreaks compared to recent previous years. In healthcare facilities, outbreaks of measles can have serious consequences because there are usually a large number of patients and HCWs exposed to it, particularly immunocompromised patients. Moreover, these are likely to have negative implications not only in terms of direct medical costs, including the cost of antibody screening tests and administrative costs, but also indirect non-medical costs, such as loss of productivity on account of reduced working hours and financial losses arising from the cancellation of healthcare services. To limit the spread of measles, early diagnosis and immediate isolation procedures, as well as the furloughing of potentially exposed HCWs who develop symptoms, remain key interventions. This study highlights the importance of assessing the prevalence of measles susceptibility among HCWs in healthcare settings, regardless of their vaccination status and age. Considering the progressive decline in antibody titers with time, measles vaccination status and laboratory evidence of immunity should be up-to-date especially among young HCWs, whether or not the HCW has documentation of receiving two-dose MMR vaccines. Furthermore, mandatory immunization should be considered for HCWs if they are identified as seronegative. This information plays a crucial role in guiding policy decisions for hospital-wide vaccination strategies aimed at preventing vaccine-preventable diseases. Further studies are necessary to determine the most cost-effective vaccination strategies for the different HCW age groups in South Korea.

## Figures and Tables

**Figure 1 vaccines-11-01505-f001:**
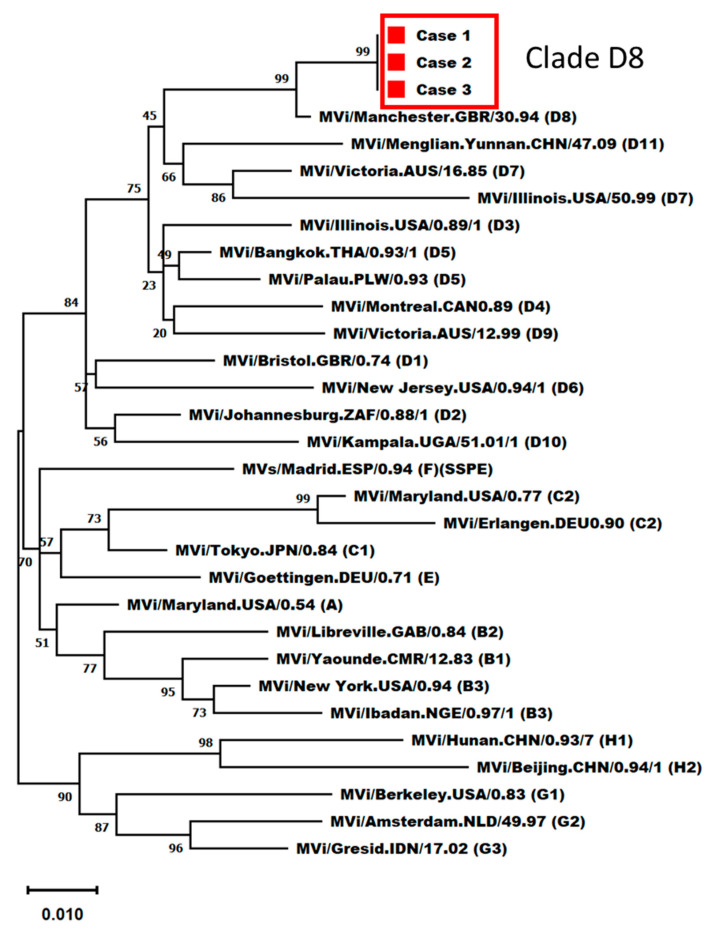
Phylogenetic analysis of the N-450 sequences.

**Figure 2 vaccines-11-01505-f002:**
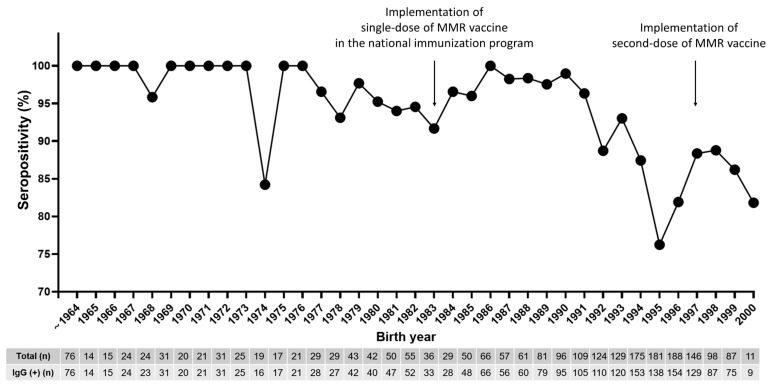
Seroprevalence of measles immunoglobulin G antibodies in healthcare workers between 2018 and 2022 stratified by year of birth. Circles denote mean seropositivity (%). HCWs, healthcare workers.

**Table 1 vaccines-11-01505-t001:** Seroprevalence of measles IgG antibodies according to the year of measles IgG testing.

	2018 *	2019 *	2020 *	2021 *	2022 *
No. of samples (*n*)	972	518	288	299	233
Seropositive (%)	926 (95.3)	450 (86.9)	257 (89.2)	271 (90.6)	211 (90.6)
Seronegative (%)	46 (4.7)	51 (9.8)	25 (8.7)	23 (7.7)	19 (8.2)
Equivocal (%)	0 (0)	17 (3.3)	6 (2.1)	5 (1.7)	3 (1.3)

* Year of measles IgG testing.

## Data Availability

In terms of the further use of our data, we ask researchers to cite our paper in their Method section. Data presented in this study are available upon request from the corresponding author.

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
