# Peer review of "A Lesson from a Measles Outbreak among Healthcare Workers in a Single Hospital in South Korea: The Importance of Knowing the Prevalence of Susceptibility"

_vaccines, 2023, doi:10.3390/vaccines11091505_

Round 1

Reviewer 1 Report

Minor editing - example line 160-161.. (..change 'we should have..' )

Author Response

. Minor editing - example line 160-161.. (..change 'we should have..' )

à As the reviewer’s advice, we have revised the manuscript as follows.

“…after Case 2 occurred, we had to evaluated measles IgG in the ER’s HCWs…”

Reviewer 2 Report

A nice study written on epidemic management and immunization, which are important issues in terms of public health.

Is there a similar situation in countries like South Korea? If an example is given, the article becomes richer.

Author Response

  1. A nice study written on epidemic management and immunization, which are important issues in terms of public health.

Is there a similar situation in countries like South Korea? If an example is given, the article becomes richer.

We are appreciated for your generous comment. In countries where measles has been either eliminated or significantly reduced, the risk of transmission may be increased among health care workers (HCWs) who do not have immunity from natural infection and experience for measles cases (Vaccine. 2012 Jun 8;30(27):3996-4001.).

 In one retrospective study conducted in 2022 in Taiwan, which declared measles elimination in 2007 subsequent to the introduction of the measles vaccine in 1978, anti-measles IgG positive rates were investigated among HCWs born before 1977 and after 1978. The seropositive rates for these two groups were found to be 94.8% and 70.2% respectively (p < 0.001), strongly associated with prior exposure to natural measles infection (BMC Infect Dis. 2022 May 4;22(1):427.).

 Additionally, in a measles outbreak among young adults in Victoria in 2002, all but one of the 22 hospitalized cases experienced delayed diagnoses due to the low index of suspicion, presenting opportunities for further transmission within the healthcare setting (Commun Dis Intell Q Rep. 2002;26(2):273-8.).

 We have amended the Discussion section of the revised manuscript as follows.

“…early diagnosis. In a measles outbreak among young adults in Victoria in 2002, all but one of the 22 hospitalized cases experienced delayed diagnoses due to the low index of suspicion, presenting opportunities for further transmission within the healthcare setting. We addressed…”

… such as waning immunity. Similar to our current study, one retrospective study conducted in 2022 in Taiwan, which declared measles elimination in 2007 subsequent to the introduction of the measles vaccine in 1978, anti-measles IgG positive rates were investigated among HCWs born before 1977 and after 1978. The seropositive rates for these two groups were found to be 94.8% and 70.2% respectively (p < 0.001), strongly associated with prior exposure to natural measles infection. Measles avidity assays…

Reviewer 3 Report

General comments

In this paper, the authors reported an in-hospital measles outbreak initiated by an imported measles case in South Korea and a following seroepidemiological study among the healthcare workers in the same hospital as one of the responses to this outbreak. The results suggested that, in addition to vaccination status and age which are considered acceptable evidence of immunity against measles by the health authorities, the susceptibility of younger healthcare workers to measles should also be emphasized in the policy-making process to prevent and control measles hospital-wide. This paper is well-written overall but does not fit in the Special Issue “Clinical Immunology: Disease Control and Prevention”. Therefore, I would suggest the authors consider submitting to other topic or journals.

Specific comments

Introduction

1.     Line 49: although measles is also called rubeola, it would be better to change it to measles virus here to keep it consistent throughout the manuscript.

Methods

2.     Line 111: I didn’t see the description of the method you used for “genetic analysis”, though you did reconstruct a phylogenetic tree in Figure 1.

Results

3.     Line 135: please rewrite this sentence as: “the PCR results indicated he was measles virus-positive …”

4.     Line 136: add ‘tested’ between ‘were’ and ‘positive’

5.     Line 138: what do you mean by “each ward did not have a door.”?

6.     Line 147: replace ‘that’ with ‘and’

7.     Line 148: add a comma before ‘both PCR’

8.     Line 159: rewrite this sentence “When the evidence was unclear, the exposed HCWs received the MMR vaccination.” I didn’t get the point.

9.     Line 169: move “and 316 patients” right after “HCWs”

10.  Line 170: I don’t think it’s a phylogeographic analysis, but a phylogenetic analysis.

11.  Line 171: you mentioned D8 genotype circulates in China, but I didn’t see that you add any D8 reference strains from China while building your phylogenetic tree.

12.  Line 172: this outbreak resulted in 3 cases and 964 potential exposures.

13.  Line 185: replace ‘had been’ with ‘were

14.  Line 190: in your Methods, you mentioned the genetic analysis targeted on a 450nt region. But, here the figure title says complete genomes of measles virus. Which one is correct?

15.  Line 198: Table 1 – the name of the first column is not “Year of test”.

See my specific comments in the report.

Reviewer 4 Report

In this study, contact tracing and seroepidemiological investigations conducted during a measles outbreak in a hospital are described. The relationship between vaccination timing, vaccine doses, and seropositivity for measles, a still significant infectious disease from a public health perspective, is also discussed. It is a well-designed and well-written valuable study. My suggestion to the authors is to replace some references, as 12 out of the 27 references used are over a decade old. I enjoyed reading this valuable study and extend my gratitude to the authors.

Author Response

  1. In this study, contact tracing and seroepidemiological investigations conducted during a measles outbreak in a hospital are described. The relationship between vaccination timing, vaccine doses, and seropositivity for measles, a still significant infectious disease from a public health perspective, is also discussed. It is a well-designed and well-written valuable study. My suggestion to the authors is to replace some references, as 12 out of the 27 references used are over a decade old. I enjoyed reading this valuable study and extend my gratitude to the authors.

We are appreciated for your generous comment. As the reviewer’s suggestion, we have edited some references as follows.

  1. Jung J, Kim SK, Kwak SH, Hong MJ, Kim SH. Seroprevalence of Measles in Healthcare Workers in South Korea. Infect Chemother. Mar 2019;51(1):58-61. doi:10.3947/ic.2019.51.1.58
  2. Avramovich E, Indenbaum V, Haber M, et al. Measles Outbreak in a Highly Vaccinated Population - Israel, July-August 2017. MMWR Morb Mortal Wkly Rep. Oct 26 2018;67(42):1186-1188. doi:10.15585/mmwr.mm6742a4

Round 2

Reviewer 3 Report

Most of my comments have been well addressed. The manuscript is good to go after minor revisions as follows:

Line 165: change 'birth' to 'born'.

Table 1: I would suggest you remove the first column name and add a note to '2018' saying 'Year of employment', as the content in first column was nothing to do with years.

Author Response

Response to Reviewer 3 Comments

Most of my comments have been well addressed. The manuscript is good to go after minor revisions as follows:

Line 165: change 'birth' to 'born'.

As the reviewer’s suggestion, we have edited the Results section of the revised manuscript as follows.

“…to exposed HCWs born after 1967 who did not…”

Table 1: I would suggest you remove the first column name and add a note to '2018' saying 'Year of employment', as the content in first column was nothing to do with years.

We totally understood the reviewer’s suggestion, and we have edited the Table 1 as follows.

Table 1. Seroprevalence of measles IgG antibodies according to the year of employment.

2018*

2019

2020

2021

2022

No. of samples (n)

972

515

288

294

234

Seropositive (%)

926 (95.3)

447 (86.8)

257 (89.2)

266 (90.5)

211 (90.2)

Seronegative (%)

46 (4.7)

51 (9.9)

25 (8.7)

23 (7.8)

20 (8.5)

Equivocal (%)

0 (0)

17 (3.3)

6 (2.1)

5 (1.7)

3 (1.3)

* Year of employment